# DNA Damage Induction Alters the Expression of Ubiquitin and SUMO Regulators in Preimplantation Stage Pig Embryos

**DOI:** 10.3390/ijms23179610

**Published:** 2022-08-25

**Authors:** Zigomar da Silva, Werner Giehl Glanzner, Luke Currin, Mariana Priotto de Macedo, Karina Gutierrez, Vanessa Guay, Paulo Bayard Dias Gonçalves, Vilceu Bordignon

**Affiliations:** 1Laboratory of Biotechnology and Animal Reproduction–BioRep, Federal University of Santa Maria, Santa Maria 97105-900, Brazil; 2Department of Animal Science, McGill University, Sainte-Anne-de-Bellevue, QC H9X 3V9, Canada

**Keywords:** swine, embryo development, ubiquitylation, SUMOylation, DNA damage

## Abstract

DNA damage in early-stage embryos impacts development and is a risk factor for segregation of altered genomes. DNA damage response (DDR) encompasses a sophisticated network of proteins involved in sensing, signaling, and repairing damage. DDR is regulated by reversible post-translational modifications including acetylation, methylation, phosphorylation, ubiquitylation, and SUMOylation. While important regulators of these processes have been characterized in somatic cells, their roles in early-stage embryos remain broadly unknown. The objective of this study was to explore how ubiquitylation and SUMOylation are involved in the regulation of early development in porcine embryos by assessing the mRNA profile of genes encoding ubiquitination (UBs), deubiquitination (DUBs), SUMOylation (SUMOs) or deSUMOylation (deSUMOs) enzymes in oocyte and embryos at different stages of development, and to evaluate if the induction of DNA damage at different stages of embryo development would alter the mRNA abundance of these genes. Pig embryos were produced by in vitro fertilization and DNA damage was induced by ultraviolet (UV) light exposure for 10 s on days 2, 4 or 7 of development. The relative mRNA abundance of most UBs, DUBs, SUMOs, and deSUMOs was higher in oocytes and early-stage embryos than in blastocysts. Transcript levels for UBs (*RNF20*, *RNF40*, *RNF114*, *RNF169*, *CUL5*, *DCAF2*, *DECAF13*, and *DDB1*), DUBs (*USP16*), and SUMOs (*CBX4*, *UBA2* and *UBC9*), were upregulated in early-stage embryos (D2 and/or D4) compared to oocytes and blastocysts. In response to UV-induced DNA damage, transcript levels of several UBs, DUBs, SUMOs, and deSUMOs decreased in D2 and D4 embryos, but increased in blastocysts. These findings revealed that transcript levels of genes encoding for important UBs, DUBs, SUMOs, and deSUMOs are regulated during early embryo development and are modulated in response to induced DNA damage. This study has also identified candidate genes controlling post-translational modifications that may have relevant roles in the regulation of normal embryo development, repair of damaged DNA, and preservation of genome stability in the pig embryo.

## 1. Introduction

Early embryo development in mammals is coordinated by regulators that are either inherited from the gametes or newly encoded by the embryo [1]. Important events for normal embryo development include chromatin remodeling, acquisition of cell totipotency, activation of transcriptional activity, and cell differentiation. Embryos gradually take control of their development during a period spanning a few cell divisions, which is referred to as the oocyte (or maternal) to embryo (or zygote) transition (OET or MZT) [2]. This is also the stage of development when the embryo genome is activated (EGA), which occurs in two phases [3]. The minor activation phase starts shortly after fertilization and involves fewer genes than the major activation phase, which represents the stage when embryo development becomes dependent on its transcriptional activity [4]. The timing of the major EGA phase is species-specific and occurs during the 4–8 cell transition in the pig embryo [5].

Another crucial component for normal embryo development is the preservation of genome integrity [6], which is needed to enable an accurate segregation of the genetic information [7]. The cell genome is constantly threatened by endogenous and exogenous genotoxic factors that can cause DNA lesions in either one, single-strand breaks (SSBs), or both, double-strand breaks (DSBs), DNA strands [8]. DSBs are the most deleterious forms of DNA damage for embryos, since they can result in genetic mutations, cell cycle arrest, cell apoptosis, and embryo mortality [6,9]. To avoid these consequences, embryos activate a DDR to repair DSBs [10]. Cells can use one of two pathways to repair DSBs, homologous recombination (HR) or non-homologous end-joining (NHEJ); however, embryos at early stages of development seem to preferentially use the HR pathway [6,11]. RAD51 and BRCA1 are important effector proteins of the HR pathway [6]. The recruitment of these effector proteins to the sites of DSBs involves the replacement of the histone H2A by its variant H2AX, which is phosphorylated at serine 139 (γH2AX or H2AX139ph) in response to DNA damage and increasingly accumulates in the chromatin near the DSB sites. This is important to anchoring the DNA repair proteins at the DSB sites and involves the modulation of chromatin accessibility and recruitment of proteins [12]. 

Ubiquitylation and SUMOylation, which refers to the reversible covalent attachment of ubiquitin and small ubiquitin-like motifs (SUMO) to proteins, are implicated in the regulation of protein structured, stability, interaction, function, and localization, and they consequently affect many cellular mechanisms, including DDR [13]. Ubiquitylation and SUMOylation usually occurs in lysine residues and involves an enzymatic cascade of either ubiquitin or SUMO activating (E1), conjugating (E2), and ligase (E3) enzymes [14,15]. Removal of ubiquitin and SUMO from proteins is catalyzed by ubiquitin-specific proteases or DUBs, and isopeptidases or SUMO-deconjugating enzymes collectively referred to as deSUMOs, in processes called deubiquitylation and deSUMOylation, respectively [16,17]. 

Several hundred genes involved in the regulation of ubiquitylation/deubiquitylation and SUMOylation/deSUMOylation have been identified as participating in the modulation of different cell processes [13], including DDR [18]. Several of these genes were shown to be expressed in embryos [19] and to participate in the modulation of developmental events such as EGA [20,21], DNA methylation [22], and cell pluripotency [23]. However, the importance of many regulators of ubiquitylation/deubiquitylation and SUMOylation/deSUMOylation for normal embryo development remains to be studied. Therefore, the goals of this study were to determine the mRNA expression profile of several genes encoding for important UBs, DUBs, SUMOs or deSUMOs in porcine embryos, and to evaluate if the mRNA expression of these genes is altered in response to the induction of DNA damage at different stages of embryo development.

## 2. Results

### 2.1. Effect of UV-Exposure on Embryo Development and DNA Damage

To validate our model of UV-induced DNA damage, embryos on days 2, 4 and 7 of development were exposed to UV for 10 s and then cultured and evaluated either 2 h later or on day 7. Blastocyst rates of embryos exposed to UV on days 2 or 4 of development were significantly lower than control embryos not treated with UV (Table 1). At all stages of development, embryos exposed to UV showed an increased fluorescent intensity for γH2A.X (Figure 1A,B). In addition, the relative mRNA abundance of genes encoding the DDR effectors *RAD51* and *KU70* was significantly decreased in D2 and D4 porcine embryos exposed to UV compared to control embryos (Figure 1C). These observations indicate the UV treatment induced DNA lesions but did not cause irreversible damage to all the embryos, given that a proportion of the UV-exposed embryos continue their development to the blastocyst stage. 

### 2.2. mRNA Levels of UBs, DUBs, SUMOs and deSUMOs during Embryo Development

The relative mRNA abundance of genes encoding UBs, DUBs, SUMOs, and deSUMOs was quantified in mature oocytes (MII stage) and embryos at developmental stages that represent pre-EGA/minor-EGA (D2), during-EGA/major-EGA (D4), and post-EGA (D7, blastocysts) stages. The relative mRNA abundance of the genes *EIF1AX* and *KDM5B* (Appendix A), which are known to be upregulated during the major-EGA stage in pig embryos [24,25], was assessed to confirm the representative stages of samples collected and used in this study. The transcript levels of most UBs (*RNF4*, *RNF8*, *RNF20*, *RNF40*, *RNF114*, *RNF126*, *RNF168*, *RNF169*, *BRCC3*, *CUL5*, *DCAF2*, *DCAF13*, *DDB1*, and *UBE2N*) (Figure 2A) and DUBs (*USP7*, *USP11*, *USP16*, *USP34*, *OTUB1*, *OTUB2* and *BAP1*) (Figure 2B) were lower in D7 blastocysts compared to either oocytes, D2 or D4 embryos. However, different mRNA expression profiles for UBs and DUBs were identified at these developmental stages. The first group comprises UBs and DUBs having a decreasing mRNA pattern during development starting either between oocytes and D2 embryos (*BRCC3* and *OTUB1*), between oocytes and D4 embryos (*RNF8* and *USP7*), or remaining relatively constant between oocytes and D4 embryos, but decreasing after (*RNF4*, *RNF126*, *RNF168*, *UBE2N*, *USP11*, *USP34*, *OTUB2* and *BAP1*) (Figure 2A,B). The second group includes UBs and DUBs that showed a transient increase in mRNA levels either on D2 (*RNF20*, *RNF40*, *RNF114*, *DCAF2*, *DDB1* and *USP16*), D2 and D4 (*DCAF13*), or D4 (*RNF169* and *CUL5*) of development (Figure 2A,B). Similar mRNA profiles were observed for genes encoding SUMO and deSUMO enzymes (Figure 3). A decreasing mRNA pattern, which remained relatively constant from oocytes to D4 embryos but was significantly lower at the blastocyst stage, was observed for the SUMOs *PIAS1*, *PIAS2* and *PIAS4* (Figure 3A), and for the deSUMOs *SENP2* and *SENP7* (Figure 3B). A transient upregulation profile in the mRNA abundance was detected for the SUMOs *UBC9* on D2 embryos, and *CBX4* and *UBA2* on D4 embryos (Figure 3A). These mRNA patterns may be indicative of genes that have roles in the regulation of oocyte maturation and during early events of embryo development, and genes that may participate in the regulation of OET and EGA mechanisms.

### 2.3. UV-Induced DNA Damage Alters mRNA Levels of UBs, DUBs, SUMOs and deSUMOs during Embryo Development

To investigate if genes encoding UBs, DUBs, SUMOs, and deSUMOs enzymes are regulated in response to DNA damage, embryos were UV-exposed for 10 s on days 2, 4 or 7 of development and mRNA was extracted 2 h after treatment to assess mRNA levels. The relative mRNA abundance of all the UBs and DUBs genes assessed in this study was altered in response to UV-treatment, as evidenced by a general reduction of mRNA levels in D2 and D4 embryos, but an increase of mRNA levels in D7 blastocysts (Figure 4 and Figure 5). On D2 of development, the relative mRNA abundance of several UBs (*RNF4*, *RNF8*, *RNF20*, *RNF114*, *RNF126*, *RNF168*, *DCAF2*, *DDB1* and *UBE2N*) (Figure 4A) and DUBs (*USP7*, *USP11*, *USP16*, *USP34*, *OUTB1*, *OUTB2* and *BAP1*) (Figure 4B), was significantly reduced in UV-treated compared to control embryos. UV treatment also resulted in a significant decrease of mRNA levels for UBs (*RNF8*, *RNF126*, *RNF168*, *BRCC3*, and *UBE2N* (Figure 4A) and DUBs (*USP7*, *USP16* and *BAP1*) (Figure 4B) in D4 embryos. On the other hand, the relative mRNA abundance of UBs (*RNF40*, *RNF168*, *BRCC3*, *CUL5*, *DCAF13*, *DDB1* and *UBE2N*) (Figure 4A) and DUBs (*USP11* and *USP34*) (Figure 4B) was significantly increased in UV-treated blastocysts compared to control blastocysts. Similarly, mRNA levels of SUMO and deSUMO genes in UV-treated embryos was differently regulated according to the embryo developmental stage. In D2 embryos, UV-induced DNA damage significantly decreased mRNA levels of SUMO (*PIAS1*, *PIAS2*, *PIAS4*, *CBX4*, and *UBC9*) (Figure 5A) and deSUMO (*SENP2*) (Figure 5B) genes compared to control embryos. In D4 embryos, UV treatment decreased the relative mRNA abundance of SUMO (*PIAS4*, *CBX4* and *UBA2*) (Figure 5A) and deSUMO (*SENP2*) (Figure 5B) genes. In D7 blastocysts, UV treatment resulted in a significant increase in the mRNA levels of SUMO (*PIAS2*, *CBX4*, *UBA2* and *UBC9*) genes compared to control blastocysts (Figure 5A). These findings reveled genes having a potential role in the regulation of DDR and preservation of genome stability in pig embryos. 

## 3. Discussion

Normal embryo development requires post-translational modifications of proteins including ubiquitylation and SUMOylation. These modifications are involved in the regulation of many developmental events such as chromatin remodeling, acquisition of cell totipotency, cell cleavage, embryo genome activation, transcription, and cell differentiation, as well as DNA repair and genome stability [13,19]. Ubiquitylation and SUMOylation are controlled by many proteins, but the expression of the genes encoding these proteins and their functions in the regulation of early embryo development of different species remain to be determined, particularly in the context of DDR. Thus, we aimed in this study to identify ubiquitylation and SUMOylation regulators that are potentially involved in the regulation of DDR during early development of pig embryos. For this, we first characterized the mRNA expression profile of 30 genes encoding important regulators of ubiquitylation, deubiquitylation, SUMOylation, and deSUMOylation in pig oocytes and embryos at different stages of development. We then used a previously tested model of DNA damage, based on a brief exposure to UV light [6,24], to demonstrate that the mRNA profiles of many of these genes are altered when DNA damage is induced. 

Analysis of mRNA profiles in control embryos that were not exposed to UV revealed two expression profiles of ubiquitylation, deubiquitylation, SUMOylation and deSUMOylation regulators. The first profile consisted of a decreasing mRNA pattern from oocyte and early-stage embryos to blastocysts, and was observed in several genes (*RNF4*, *RNF8*, *RNF126*, *RNF168*, *BRCC3*, *UBE2N, USP7*, *USP11*, *USP34*, *OTUB1*, *OTUB2, BAP1, PIAS1*, *PIAS2, PIAS4, SENP2*, and *SENP7*). This profile indicates that the transcripts of these genes are mainly maternally derived and stored in the mature oocyte, suggesting these genes may participate in the regulation of oocyte or early embryo stage functions, such as DNA repair and stability. For instance, RNF4 was shown to affect germ cell development and oocyte maturation [26], USP7 contributes to the maintenance of DNA methylation [27], and down-regulation of PIAS4 is necessary for normal EGA in mouse embryos because it is a negative regulator of DPPA2 [28]. In addition, overexpression of SENP2 affected spindle formation in MII oocytes [29], and the suppression of SENP2 caused trophoblast defects and mortality of mouse embryos [30]. Moreover, meiotic arrest, defective maternal-to-zygotic transition, and embryo death were observed in mice as a consequence of SENP7 suppression [31]. SENP7 has also been related to a negative regulation of HP1 (also known as Chromobox 5-CBX5), which interacts with H3K9me3 to repress gene transcription through the establishment of heterochromatin [32]. 

The second profile consisted of genes having a transient upregulation in mRNA levels on D2 or D4 of development (*RNF20*, *RNF40*, *RNF114*, RNF169, *CUL5*, *DCAF2*, *DCAF13*, *DDB1, USP16*, *CBX4*, *UBA2*, and *UBC9*). These developmental stages correspond to the minor and major EGA phases in porcine embryos, which suggests these genes may be involved in the regulation of OET in pigs. In support of this are findings from previous studies in other species, for example, RNF114 participates in the regulation of EGA in mouse embryos by ubiquitinating the TGF-beta activated kinase 1 and MAP3K7-binding protein 1 (TAB1) [20], and the CBX5 [33], targeting them to degradation. TAB1 degradation activates the NF-kB pathway, which promotes maternal mRNA clearance in mouse embryos [20], while CBX5 degradation allows gene transcription through its interaction with dimethylated (H3K9me2) and trimethylated (H3K9me3) histone H3 lysine 9, which represses gene transcription [20,33]. On the other hand, DCAF13 was shown to negatively regulate the histone lysine methyltransferase SUV39H1, which leads to a decrease of H3K9me3 in mouse embryos [34]. Moreover, USP16 was shown to be necessary for EGA and subsequent development of mouse embryos through its role as a deubiquitinase for H2AK119ub1 [35]. In addition, RNF20, which is part of the RNF20/RNF40 complex, may also affect EGA through the regulation of ZSCAN4 [36], which is known to regulate EGA in mouse embryos [37]. Although a specific role of other genes involved in the regulation of EGA has not been reported, CUL5 was shown to be necessary for the normal development of mouse embryos [38]. Furthermore, the suppression of DCAF2 in mouse oocytes resulted in embryo arrest at the 1–2 cell stage due to the accumulation of DNA damage [39]. It has also been shown that deletion of DDB1 in mouse oocytes decreased their viability by affecting TET dioxygenases [40], and delaying meiotic resumption [41]. Moreover, embryos deficient in UBC9 died during early development due to defects in chromosome segregation [42]. In addition, supplementation of the UBC9 protein in the culture medium had a detrimental effect on oocyte maturation and embryo development in pigs [43]. On the other hand, UBA2 supplementation in the culture medium increased both oocyte maturation and development of pig embryos [44]. 

In the context of DDR after UV treatment, we observed an overall trend for decreasing gene transcripts when DNA damage was induced either during minor EGA (D2 embryos) or major EGA (D4 embryos) stages. However, an overall trend for increasing gene transcripts was observed when DNA damage was induced in D7 blastocysts. Indeed, the transcript levels of 22 (*RNF4*, *RNF8*, *RNF20*, *RNF114*, *RNF126*, *RNF168*, *DCAF2*, *DDB1*, *UBE2N*, *USP7*, *USP11*, *USP16*, *USP34*, *OTUB1*, *OTUB32*, *BAP1*, *PIAS1*, *PIAS2*, *PIAS4*, *CBX4*, *UBC9*, and *SENP2*) and 12 (*RNF8*, *RNF126*, *RNF168*, *BRCC3*, *UBE2N*, *USP7*, *USP16*, *BAP1*, *PIAS1*, *CBX4*, *UBA2*, and *SENP2*) genes were significantly decreased in UV-treated embryos compared to a control on D2 and D4 of development, respectively. We anticipate this decrease reflects an accelerated translation of mRNA transcripts as part of the DDR response, which were still maternally derived without being compensated by newly produced transcripts due to the rather limited transcriptional activity of pig embryos at those stages of development [5,24]. Nonetheless, we did not assess if the levels of proteins encoded by these transcripts increased during DDR. On the other hand, UV treatment of blastocyst, which are fully transcriptionally competent, resulted in the upregulation of 13 (*RNF40*, *RNF168*, *BRCC3*, *CUL5*, *DCAF13*, *DDB1*, *UBE2N*, *USP11*, *USP34*, *PIAS2*, *CBX4*, *UBA2*, and *UBC9*) out of 30 genes, which suggests an increased demand of these genes for the regulation of DDR in pig blastocysts. 

Findings from this study also suggest that both ubiquitylation and SUMOylation are involved in the regulation of the HR pathways of DNA repair, which is preferentially used to repair DSBs in early developing pig embryos [6]. We observed that mRNA levels of *RNF8* and *RNF168*, which are core components of the ubiquitin-mediated response to DSBs [45,46], were regulated in response to UV treatment. The ubiquitin-mediated DDR is initiated by RNF168, which promotes monoubiquitination of H2A/H2AX at K13-15 and is continued by RNF8, which extends the ubiquitin chain by polyubiquitination of H2A/H2AX at the lysine 63 [47]. These changes trigger NHEJ factors to DBS foci, including 53BP1. However, RNF169 promotes the recruitment of BRCA1 and RAD51 to DSB foci, which favors the HR pathway [46]. We observed that *RNF168* mRNA fluctuated in UV-treated embryos, despite not being statistically different, and *RAD51* mRNA was altered in response to UV treatment, agreeing with findings of previous studies [6,48].

Findings from this study also revealed that mRNA levels of several other genes with known roles in the regulation of the DNA repair by the HR pathways were altered in the UV-treated embryos. These include genes encoding enzymes involved in the regulation of chromatin accessibility (*RNF20, RNF40*, *DCAF2*, *RNF114*, *DDB1*, *UBE2N, USP7, USP11,* and *USP16* [49,50]), recruitment of HR effectors (*RNF4*, *RNF126*, *BRCC3*, *UBE2N*, *CUL5, USP16, USP34*, *BAP1*, *PIAS*, *PIAS4, CBX4, UBC9*, and *SENP2* [51,52,53]), replacement of MDC1 and RPA1 by BRCA2 and RAD51 in DSB sites (*RNF4* [54]), expression of HR effectors (*RNF126* and *OTUB2* [55,56]), degradation of NHEJ effectors (*RNF4* [57]), clearance of HR effectors after DNA repair (*OTUB1*, *BRCC3*, *SENP2* [53,58]), and regulation of RNF8/RNF168 levels (*OTUB2* [59]). Our observations suggest these genes are potentially involved in the HR repair pathway in pig embryos, but their importance and specific roles require further investigation.

## 4. Material and Methods

Unless otherwise indicated, all chemicals and reagents were purchased from Sigma-Aldrich Canada Co. (Oakville, ON, Canada).

### 4.1. In Vitro Embryo Production

Porcine ovaries were obtained from a local abattoir (CBCo Alliance Inc., Les Cèdres, QC, Canada) and transported to the laboratory at 32–35 °C in 0.9% saline solution containing penicillin (100 UI/mL) and streptomycin (10 mg/mL). Cumulus-oocyte complexes (COCs) were collected by aspirating 3–6 mm follicles using a 20-gauge needle attached to a 10 mL syringe. Only COCs with at least three layers of cumulus cells and homogeneous granulated cytoplasm were selected for in vitro maturation (IVM). Groups of 30 COCs were matured at 38.5 °C in 5% CO_2_ and 95% air for 22 h in 90 µL of maturation medium consisting of TCM-199 (Life technologies, Burlington, ON, Canada), supplemented with 20% of porcine follicular fluid, 1 mM dibutyryl cyclic adenosine monophosphate (dbcAMP), 0.1 mg/mL cysteine, 10 ng/mL epidermal growth factor (EGF; Life technologies), 0.91 mM sodium pyruvate, 3.05 mM D-glucose, 5 IU/mL hCG (Intervet Canada Corp, Kirkland, QC, Canada), 10 µg/mL FSH (Vetoquinol, Lavaltrie, QC, Canada), and 20 µg/mL gentamicin (Life technologies). COCs were transferred to the same IVM medium, but without hCG, FSH, and dbcAMP, for an additional 20 to 22 h under the same conditions. 

For in vitro fertilization (IVF), cumulus cells were removed after 44 h of IVM by vortexing COCs in TCM199 HEPES-buffered medium supplemented with 0.1% hyaluronidase. Selected oocytes were washed and transferred to 90µL drops of porcine TBM-Fert, consisting of Tris-buffered media supplemented with 2 mg/mL bovine serum albumin (BSA), 2 mM caffeine, and 20 µg/mL gentamicin. Semen from fertile boars was prepared by washing in TBM-Fert devoid of caffeine and then resuspending in regular TBM-Fert, immediately before coincubation with the mature oocytes. Approximately 20,000 motile sperm diluted in 10 µL of TBM-Fert were added to each 90 µL drop and incubated with the oocytes for 5 h. Embryos were cultured in groups of 30 in 60 μL of porcine zygote medium 3 (PZM-3), supplemented with 3 mg/mL BSA, and 5 mM hypotaurine. On day 5 of culture, medium was supplemented with 10% fetal bovine serum (FBS; Thermo Fisher Scientific, Waltham, MA, USA). Embryos were cultured under mineral oil at 5% CO_2_, 95% air and 38.5 °C for 7 days or until they were collected for analysis.

### 4.2. Ultraviolet Treatment 

To induce DNA damage, embryos on days 2, 4 or 7 of development were placed in a 35 mm cell culture dish containing 1 mL of PZM-3 medium, and then transferred to a warming plate at 38.5 °C placed inside a biologic safety cabinet (1300 Series A2, Thermo Fisher Scientific). The UV source consisted of a 30W UV-C lamp with wavelength of 253.7 nm (Philips, Eindhoven, The Netherlands), which was located 63 cm away from culture dish containing the embryos. The lid of the culture dish was removed, the UV light was turned on, and embryos were exposed once for 10 s, based on previous UV dose-response results performed in our laboratory [6,24,48]. Control embryos were submitted to the same procedure but were not exposed to UV light. Control and UV-treated embryos were replaced in culture for 2 h and were then collected for mRNA extraction or fixed to determine the incidence of DSBs.

### 4.3. RNA Extraction and Quantitative Real-Time PCR (qPCR)

Samples for RNA extraction consisted of either 10 oocytes, 10 D2 or D4 embryos, or 5 blastocysts per treatment and replicate, and the experiment was repeated three times. Total RNA was extracted by using the PicoPure™ RNA Isolation kit (Thermo Fisher Scientific). RNA was treated with DNAse I (RNase-Free DNase Set; Qiagen, Louiville, KY, USA) to eliminate any potential contamination of genomic DNA, and then reverse transcribed by using the Superscript^®^ VILO™ cDNA Synthesis Kit (Life Technologies). Total cDNA (20 μL) was diluted 1:10 in water and quantitative real-time PCR reactions containing 2 μL of the diluted sample cDNA, 250 nM of appropriate primers (Table 2) and the Advanced qPCR Mastermix (Wisent Bio Products, Montreal, QC, Canada), in a total volume of 10 μL reaction, were performed by using a CFX Connect™ Real-Time PCR Detection System (Bio-Rad, Hercules, CA, USA). Thermocycler parameters were 5 min at 95 °C, followed by 40 cycles of 95 °C for 15 s and 60 °C for 30 s, and ending with a dissociation curve analysis. Samples were run in duplicates and the standard curve method was used to determine the relative abundance of mRNA for each gene. Relative mRNA abundance was normalized to the mean abundance of the internal control gene *H2A*. All reactions had efficiency between 90% and 110%, r^2^ ≥ 0.98 and slope values between −3.6 to −3.1. Dissociation curve analyses were performed to validate the specificity of the amplified products. Primers were designed by using the Primer-Blast tool (https://www.ncbi.nlm.nih.gov/tools/primer, accessed on 1 March 2022), or selected based on previous publications, and synthesized by IDT (Windsor, ON, CA). The relative mRNA abundance of genes involved in DNA damage repair (*RAD51*, *BRCA1*, *KU70* and *KU80*), embryonic genome activation (*EIF1AX* and *KDM5B*); ubiquitylation (*RNF4*, *RNF8*, *RNF20*, *RNF40*, *RNF114*, *RNF126*, *RNF168*, *RNF169*, *BRCC3*, *CUL5*, *DCAF2*, *DCAF13*, *DDB1* and *UBE2N*), deubiquitylation (*USP4*, *USP7*, *USP11*, *USP16*, *USP34*, *OTUB1*, *OTUB2*, *BAP1*), SUMOylation (*PIAS1*, *PIAS2*, *PIAS4*, *CBX4*, *UBA2*, *UBC9*), and deSUMOylation (*SENP2* and *SENP7*) was assessed in samples of control and UV-exposed embryos at different stages of development.

### 4.4. Immunofluorescence Detection of DNA Breaks

Embryos were fixed for 15 min in 4% paraformaldehyde, transferred to PBS supplemented with 0.3% BSA, and stored at 4 °C. Embryos were incubated for 4 h at 37 °C with permeabilization solution (PBS supplemented with 0.3% BSA and 1% Triton X-100^®^), and then for 1 h in blocking solution (PBS supplemented 3% BSA and 0.2% Tween-20). Embryos were then incubated overnight with the primary antibody, mouse monoclonal anti-phospho-histone H2A.X (Ser139; EMD Millipore, 05-636) diluted 1:500 in blocking solution. Negative control samples were incubated in blocking solution overnight in the absence of the primary antibody. Embryos were then rinsed 3 times for 30 min each in blocking solution before incubation in the dark for 1 h at room temperature in the presence of the secondary antibody, anti-mouse IgG Cy3-conjugated (Jackson ImmunoResearch Laboratories, West Grove, PA, USA; 1:1000). Embryos were then rinsed in blocking solution for 30 min, incubated for 20 min in 10 μg/mL 4,6-Diamidino-2-phenylindole, dilactate (DAPI), and rinsed once more in blocking solution for 30 min before mounting on slides using Mowiol. Slides were examined using a Nikon Eclipse 80i microscope (Nikon Instruments Inc., Melville, NY, USA), and images were captured at 40× (D2 and D4 embryos) or 20× (D7 embryos) magnification using a Retiga 2000R monochrome digital camera (QImaging, Surrey, BC, Canada). The exposure gains and rates were consistent between samples. Occurrence of DSBs was determined based on the immunofluorescence intensity of γH2A.X on each cell nucleus for D2 (control n = 21 and UV n = 18) and D4 (control n = 22 and UV n = 20) embryos, or on the entire blastocyst area for D7 embryos (control n = 5 and UV n = 5), by using the SimplePCI imaging software [6]. 

### 4.5. Statistical Analysis

Data were obtained from a minimum of three replicates for each experiment. All statistical analyses were performed by using the JMP15 software (SAS Institute, Cary, NC, USA). Data were tested for normality by using the Shapiro–Wilk test and were normalized when necessary. ANOVA, followed by Student’s *t*-test, was used for mean comparisons between two groups, and Least Square Means followed by Tukey HSD when more than two groups were compared. A *p* < 0.05 value was considered to indicate a significant difference between groups. 

## 5. Conclusions

Findings in this study highlight the mRNA profile of genes encoding important regulators of ubiquitylation, deubiquitylation, SUMOylation and deSUMOylation during early development of pig embryos. This study identified genes that are transcribed around the period when the embryo genome is activated, which suggests they may be involved in the process. We have also identified genes whose transcripts are altered in response to DNA damage, which suggests their potential role in the regulation of DDR and the preservation of genome stability in porcine embryos. Further functional characterization of these genes is required given their potential impacts on fertility, genome stability, and efficiency of embryo technologies.

## Figures and Tables

**Figure 1 ijms-23-09610-f001:**
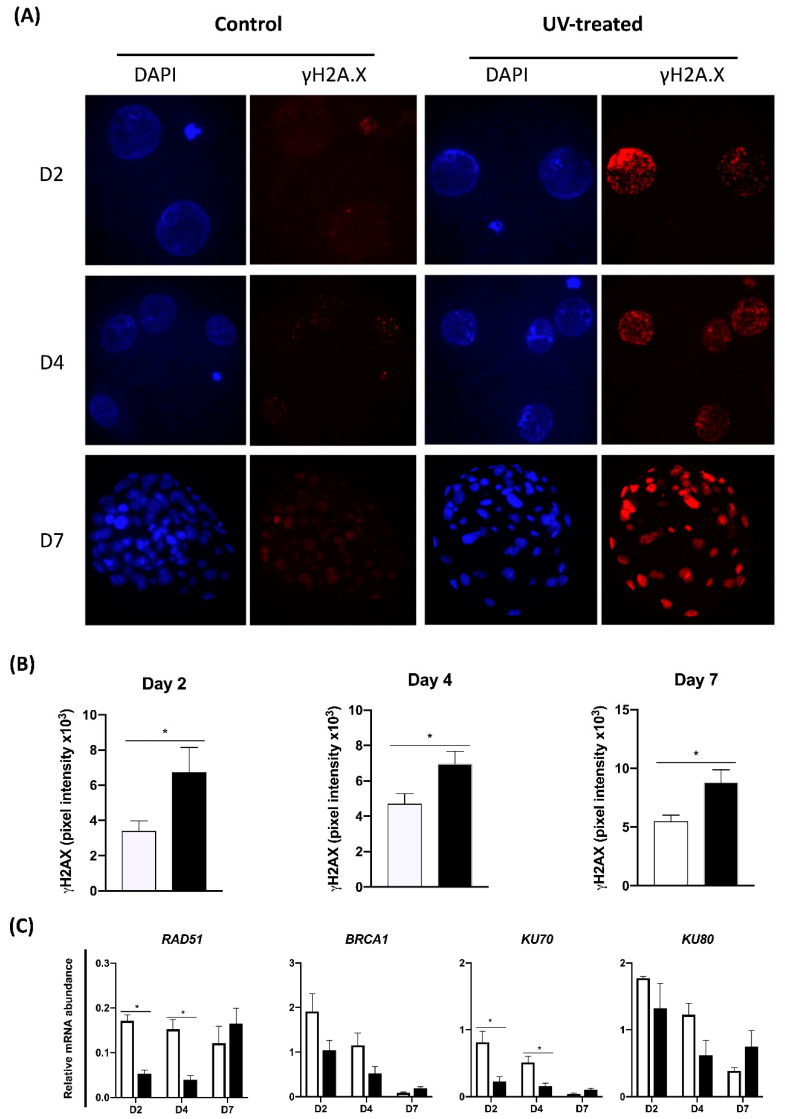
Effect of UV treatment for 10 s on DNA damage and expression of DNA repair regulators. (**A**) Representative images of control and UV-treated embryo at D2, D4 and D7 of development. Blue, cell nuclei stained with DAPI. Red, immunofluorescence signal of γH2A.X indicating increased DNA damage in UV-treated embryos. (**B**) Quantification of immunofluorescent signal (Pixel intensity, arbitrary units) on the cell nuclear area (D2 and D4) or the entire embryo area (D7) in control and UV-treated embryos assessed 2 h after UV exposure. (**C**) Relative mRNA abundance of genes involved in the regulation of DNA damage repair by the HR (RAD51, BRCA1) and NHEJ (KU70, KU80) repair pathways. ◻︎ Control embryos; ◼︎ UV-treated embryos. Asterisks indicate statistical differences (*p* < 0.05) between control and UV-treated embryos.

**Figure 2 ijms-23-09610-f002:**
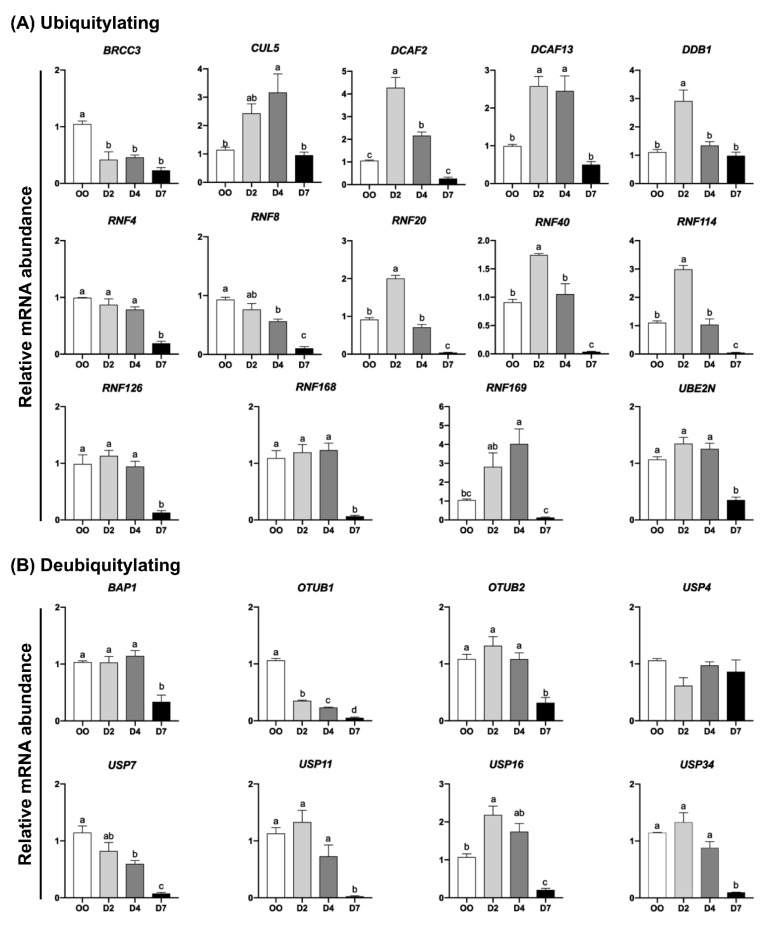
Relative mRNA abundance of genes encoding ubiquitylation (**A**) and deubiquitylation (**B**) enzymes in porcine oocytes and embryos at different stages of development. OO, MII oocytes; D2, day 2 embryos; D4, day 4 embryos; D7, day 7 blastocysts. Different letters indicate statistical differences (*p* < 0.05) between developmental stages.

**Figure 3 ijms-23-09610-f003:**
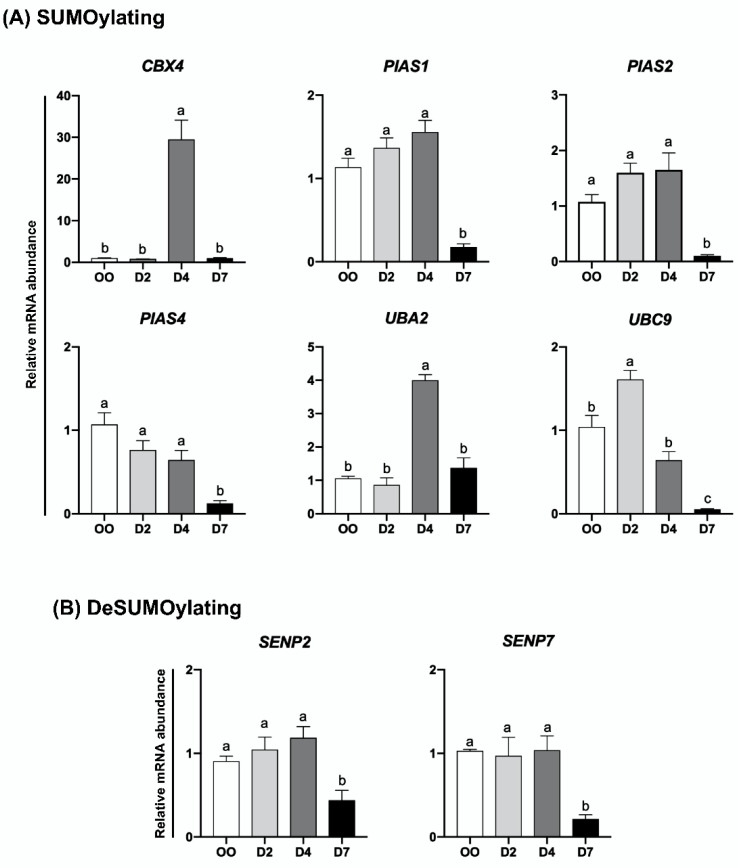
Relative mRNA abundance of genes encoding SUMOylation (**A**) and deSUMOylation (**B**) enzymes in porcine oocytes and embryos at different stages of development. OO, MII oocytes; D2, day 2 embryos; D4, day 4 embryos; D7, day 7 blastocysts. Different letters indicate statistical differences (*p* < 0.05) between developmental stages.

**Figure 4 ijms-23-09610-f004:**
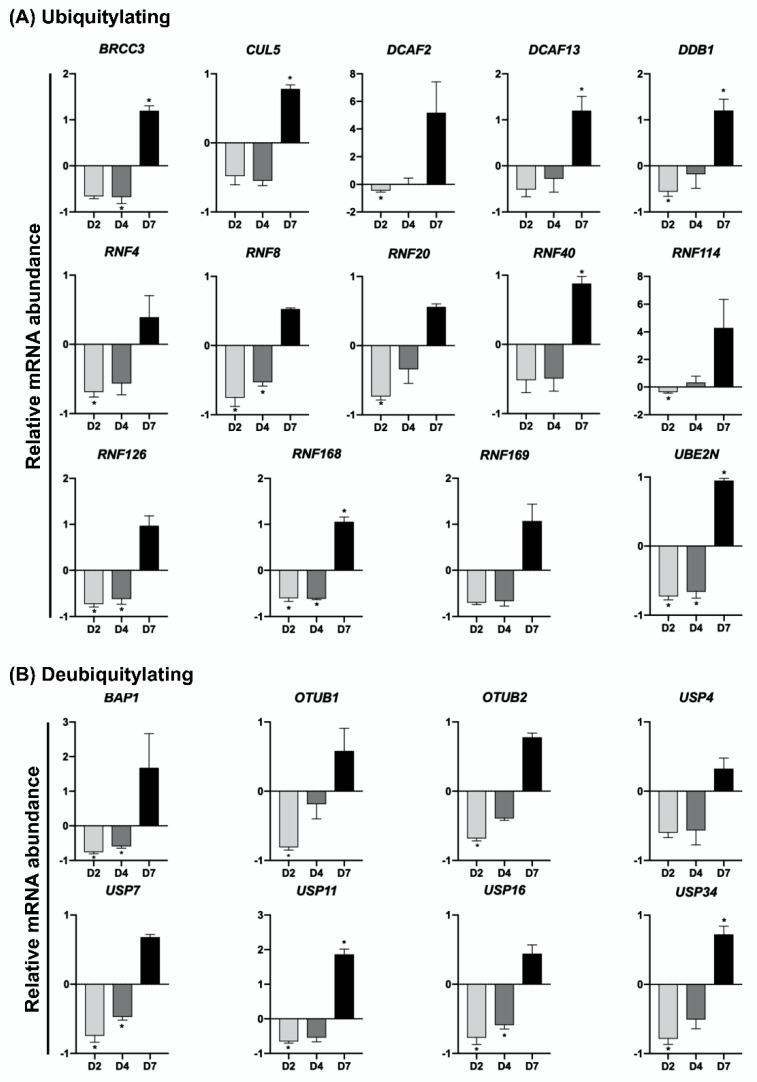
Relative mRNA abundance of genes encoding ubiquitylation (**A**) and deubiquitylation (**B**) enzymes in porcine embryos treated with UV at different stages of development. The abundance of transcripts is presented in relation to the levels observed in control embryos (not treated with UV) at the same developmental stage. D2, day 2 embryos; D4, day 4 embryos; D7, day 7 blastocysts. Asterisks indicate statistical differences (*p* < 0.05) between control and UV-treated embryos.

**Figure 5 ijms-23-09610-f005:**
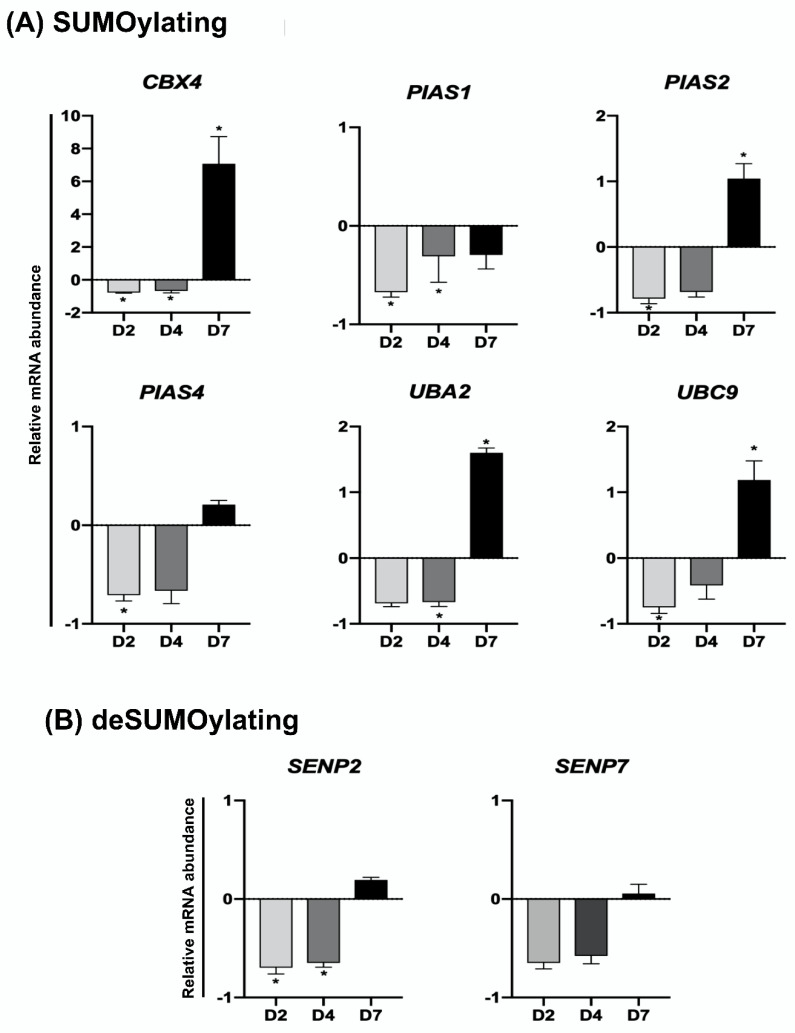
Relative mRNA abundance of genes encoding SUMOylation (**A**) and deSUMOylation (**B**) enzymes in porcine embryos treated with UV at different stages of development. The abundance of transcripts is presented in relation to the levels observed in control embryos (not treated with UV) at the same developmental stage. D2, day 2 embryos; D4, day 4 embryos; D7, day 7 blastocysts. Asterisks indicate statistical differences (*p* < 0.05) between control and UV-treated embryos.

**Table 1 ijms-23-09610-t001:** Development of embryos that were exposed to UV light for 10 s.

Stage of Development	UV Exposure	Cultured Embryos	Blastocyst n (%)
Day 2	−	103	22 (21.36) ^a^
	+	126	6 (4.76) ^b^
Day 4	−	150	33 (22) ^a^
	+	156	7 (4.49) ^b^
Day 7 (blastocysts)	−	39	39 (100)
	+	39	39 (100)

At total of 1920 COCs were in vitro matured and fertilized in four different replicates to produce the embryos for this experiment. Different letters indicate statistical difference (*p* < 0.05).

**Table 2 ijms-23-09610-t002:** Primers used in the quantitative real-time PCR reactions.

Gene Name	Forward (5′ to 3′)	Reverse (5′ to 3′)	Reference or Accession Number
BRCC3	TTGAGCTGCCCAAGATCCTC	CGGACATCTGACTGCACAGA	a
CUL5	CCCAAAGCTCAAACGGCAAG	AGTGAGCTGTAAGCGTCCAA	b
DCAF2	GCTGCCTATATCTGGAAGGTCT	ACCTCTTGCGAATGACCCAG	XM_013979970.2
DCAF13	TTCCTTGCTTCCCTGGATGG	CGTACAAAACCTTCATGCGC	XM_005662912.3
DDB1	GAACGAAAGACTGAACCGGC	GCCATCGTCGTACTGTAGGT	XM_003122651.6
RNF4	GTGGACCTGACGCACAATGA	CGTCATCACTGCTCACCACA	c
RNF8	AGAGGACCTGAAGCAACAGC	GTCCTTCCTGCTGCGGTTTA	d
RNF20	ACATTGGCACAGGGGAGAAG	TGCCTCCATTGCCTTACGTT	XM_001926594
RNF40	CCTTCAGGAGAAGCATCGCA	ACAGACACTCTTGACTCCGC	e
RNF114	GTCCATAGCACGGACACCAA	CCGACGCTGGATGTGTTCTA	NM_001001869.1
RNF126	CTTGGATGCCATCATCACGC	GGAGGGCCTGGATTTTCTCTT	f
RNF168	GTGTCTTGGCATTGCCCTTG	CAGACTGTGATGCTGACCCA	g
RNF169	GTTGTCCTGCACGTCTCTCA	CCGACCGGTGTGTCTGTATC	h
UBE2N	GCCGTACTCGGGATTTGACA	GAGCGTTGCTCTCATCTGGT	XM_005664262.3
USP4	TAAATACGATGCCGGGGAGC	GCTGTCGAAGCCCACATACT	i
USP7	TAAATACGATGCCGGGGAGC	GTTTTGGTCCGTCTGAGGGT	j
USP11	ATTCGCTCGACCTCGTCTTA	TCTGCGTTCTTGATCCACAG	XM_003484105.2
USP16	GCCAAGATTGTAAGACGGACA	AGCCCTGATGGCCACATTTA	k
USP34	GCAAGTTTGTTGCTGCTGGA	TGCCACACAGTCCAGGATTC	l
OTUB1	AGTATGCCGAGGACGACAAC	TGGTCTTGCGGATGTACGAG	NM_001162403.1
OTUB2	TCAGCCTTCATCAGGAACCG	GGCTGTGATCTGGATGTGGT	XM_001925368.4
BAP1	TTCAGCCCTGAGAGCAAAGG	GGGCCTGGCATGGCTATTAT	m
PIAS1	TCAGCTCTTCACCCAGTCCA	AGCGCTGACTGTTGTCTGAT	XM_003121749.6
PIAS2	CACAAAGCAGCCCAACCAAA	ATGAAGGCGGAATAGCAGCA	n
PIAS4	AGAGCGGACTCAAACACGAA	GGTGTGGGTTCTGAGCTCTT	XM_003354007.4
CBX4	ATCGCCTTCCAGAACAGGGAA	ATTGGAACGACGCGCAAAG	XM_003131145.4
UBA2	GGAGCCGACTTCAAGCAGAT	TGGGGCATCACCAACAACTT	XM_021097539.1
UBC9	CAAGCAGAGGCCTACACGAT	AAGGTCGCTGCTTATGAGGG	o
SENP2	TTCTGAAGAGGGTGGCAAGG	GGAGGAACGGAGTTTCCATGA	p
SENP7	CCCATTTCAAGTGTCCCTGC	AGGCAACCCAAAGAAAGAGGA	q
RAD51	CGGTGGAAGAGGAGAGCTTTG	TTTAGCTGCCTCGGTCAGAAT	[24]
BRCA1	CTTCTGTGGTGAAGGACCCC	TCACATGGAAGCCACTGTCC	[24]
KU70	TTCAAGCCCTTGGGAATGCT	CTTGGTGAGCAGAGCAGTGA	[24]
KU80	TTCCTGAGAGCCCTTCGAGA	TTTGGGCTTCCTCGACTGTG	[24]
EIF1AX	ACACCTCCCCGATAGGAGTC	TTGAGCACACTCTTGCCCAT	[24]
KDM5B	GACGTGTGCCAGTTTTGGAC	TCGAGGACACAGCACCTCTA	[24]
H2A	GGTGCTGGAGTATCTGACCG	GTTGAGCTCTTCGTCGTTGC	[24]

(a) Homologous region among three transcripts: XM_021079471.1, XM_021079472.1 and XM_021079474.1; (b) Homologous region among three transcripts: XM_005667299.3, XM_021062739.1 and XM_005667298.3; (c) Homologous region among three transcripts: NM_001044528, XM_005666483 and XM_005666483; (d) Homologous region among six transcripts: XM_005665930, XM_013977873, XM_005665931, XM_013977872, XM_005665932 and XM_021098656; (e) Homologous region among two transcripts: XM_021086412 and XM_021086413; (f) Homologous region among two transcripts: XM_021084297 and XM_021084298; (g) Homologous region among four transcripts: XM_021070159.1, XM_021070157.1, XM_021070160.1 and XM_021070158.1; (h) Homologous region among six transcripts: XM_021062504.1, XM_021062508.1, XM_021062509.1, XM_021062505.1, XM_021062506.1 and XM_021062507.1; (i) Homologous region among eight transcripts: NM_001243188.1, XM_013981642.2, XM_005669506.3, XM_013981641.2, XM_021068408.1, XM_005669505.3, XM_021068409.1 and XM_013981644.2; (j) Homologous region among two transcripts: NM_001135680.1 and XM_021085575.1; (k) Homologous region between twenty transcripts: XM_005670321.3, XM_021070872.1, XM_021070870.1, XM_021070871.1, XM_021070876.1, XM_005670319.3, XM_005670311.3, XM_005670318.3, XM_021070875.1, XM_005670313.3, XM_021070877.1, XM_005670314.3, XM_005670320.3, XM_005670317.3, XM_021070879.1, XM_003358897.4, XM_005670312.3, XM_021070873.1, XM_021070878.1 and XM_005670316.3; (l) Homologous region between thirteen transcripts: XM_021087489.1, XM_013996121.2, XM_021087493.1, XM_021087492.1, XM_021087495.1, XM_021087491.1, XM_021087490.1, XM_021087499.1, XM_021087500.1, XM_021087496.1, XM_021087498.1, XM_021087501.1 and XM_021087497.1; (m) Homologous region between two transcripts: XM_005669632.3 and XM_001925236.6; (n) Homologous region between eight transcripts: XM_021092649.1, XM_021092655.1, XM_021092679.1, XM_021092622.1, XM_021092632.1, XM_021092658.1, XM_021092626.1 and XM_021092641.1; (o) Homologous region between six transcripts: XM_005655133.3, XM_021085705.1, XM_021085704.1. XM_005655131.3, XM_021085707.1 and XM_021085706.1; (p) Homologous region among eight transcripts: XM_021069989.1, XM_021069991.1, XM_021069992.1, XM_021069992.1, XM_021069994.1, XM_021069996.1, XM_021069993.1 and XM_021069995.1; (q) Homologous region among four transcripts: XM_005657117.3, XM_013990162.2, XM_013990163.2, XM_013990164.2.

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
