# Peer review of "DNA Damage Induction Alters the Expression of Ubiquitin and SUMO Regulators in Preimplantation Stage Pig Embryos"

_ijms, 2022, doi:10.3390/ijms23179610_

Round 1

Reviewer 1 Report

The manuscript entitled “DNA damage induction alters the expression of ubiquitin and SUMO regulators in preimplantation stage pig embryos” by Silva et al. is a descriptive study addressing the transcriptional profiles of  genes related to ubiquitination, deubiquitination, SUMOylation, or deSUMOylation enzymes during early development of pig embryos and UV-induced DNA damage. For the authors merit, they have used a model previous stablished by their own group (10 sec UV exposure to induced DNA damage) in order to investigate potentially regulated pathways of the embryonic DNA damage repair during preimplantation period. They concluded that they identified candidate genes controlling post-translational modifications associated with normal embryonic development and repair of DNA damage. The manuscript is well written and provided important information to the field.

Major:

1-It was not clear in the manuscript how the authors addressed the natural differences in total RNA content found between different developmental stages in order to investigate transcriptional profiles during early embryo development. Did you use cDNA equivalent to one embryo?

2-Despite it was a previous described model to induce DNA damage through embryo UV exposure and checked by H2A.x expression (using immunofluorescence assay), it will be valuable if the authors could provide the extent/consequence of DNA damage by another molecular assay (e.g., apoptosis or necrosis assay) in relation to the total number of embryonic cells.

Minor

L191-2 and Table 2: provide number of COCs 

L146: provide the exact/mean number of cells for each development stage (day 2 and 4, specially)

L151: include “once” before “…for 10 s,…”

L158-59: Improve the description of the numbers of embryos or oocytes per pool. What was the methodological decision to use these numbers per pool for each stage? How many biological replicate was evaluated?

L163-169: improve general and specific description of the RT-qPCR conditions (volume of cDNA per reaction, primers concentration, temperature, cycles, qPCR final reaction volume…)

L181 provide the numbers of each embryo development stage used

Figure 2: provide scale bar and y axis identification and unit of the mRNA abundance (do the same for all the other mRNA abundance level figures)

Author Response

Comments and Suggestions for Authors

The manuscript entitled “DNA damage induction alters the expression of ubiquitin and SUMO regulators in preimplantation stage pig embryos” by Silva et al. is a descriptive study addressing the transcriptional profiles of  genes related to ubiquitination, deubiquitination, SUMOylation, or deSUMOylation enzymes during early development of pig embryos and UV-induced DNA damage. For the authors merit, they have used a model previous stablished by their own group (10 sec UV exposure to induced DNA damage) in order to investigate potentially regulated pathways of the embryonic DNA damage repair during preimplantation period. They concluded that they identified candidate genes controlling post-translational modifications associated with normal embryonic development and repair of DNA damage. The manuscript is well written and provided important information to the field.

R: We thank the reviewer for highlighting the relevance our findings and for the suggestions to improve the manuscript.

Major:

1-It was not clear in the manuscript how the authors addressed the natural differences in total RNA content found between different developmental stages in order to investigate transcriptional profiles during early embryo development. Did you use cDNA equivalent to one embryo?

R: As explained in the materials and methods, the quantification of mRNA abundance is expressed in relation to the mRNA abundance of the internal control (housekeeping) gene, which is the histone H2A. Thus, the quantification indicates the mRNA abundance of each gene in relation to the mRNA abundance of the H2A at each stage of development or treatment. Assuming that housekeeping genes have no or very minor variations between cells, as it has been  demonstrated by many studies including in embryos, this method of quantification not only enables comparisons between embryos at different stages of development or different treatments, but it also corrects any potential variations that may occur between samples during RNA extraction or cDNA production. This is the standard procedure used to quantify and compare transcript levels between samples of embryos at different stages of develop or from different treatments at the same stage of development.  

2-Despite it was a previous described model to induce DNA damage through embryo UV exposure and checked by H2A.x expression (using immunofluorescence assay), it will be valuable if the authors could provide the extent/consequence of DNA damage by another molecular assay (e.g., apoptosis or necrosis assay) in relation to the total number of embryonic cells.

R: we thank the Reviewer for the suggestion. In fact, we have previously evaluated the rate of apoptosis, as assessed by the number of cleaved caspase 3 positive cells, and observed that the rate of apoptotic cells was not significantly increased in embryos that developed to the blastocyst stage after they were exposed to UV for 10 seconds (Dicks et al. Mol Reprod Dev. 2019;1–13. DOI: 10.1002/mrd.23305).

It is worth highlighting that our gene expression analyses and quantification of the immunofluorescence intensity for gH2AX were performed only 2h after UV exposure. Therefore, it is very unlikely that cells were undergoing apoptosis when samples were collected for mRNA or gH2AX analysis.

Minor

L191-2 and Table 2: provide number of COCs

R: The number of COCs was included in the caption of the Table 2, as suggested. 

L146: provide the exact/mean number of cells for each development stage (day 2 and 4, specially)

R: The total number of cells used to quantify gH2A.X on D2 and D4 embryos has been included as suggested.

L151: include “once” before “…for 10 s,…”

R: Included as suggested.

L158-59: Improve the description of the numbers of embryos or oocytes per pool. What was the methodological decision to use these numbers per pool for each stage? How many biological replicate was evaluated?

R: The number of embryos per pool was planned to obtain enough mRNA/cDNA to quantify many gene transcripts, as it was required for this study. The number was based on many previous studies conducted in our lab using porcine embryos.  

Embryos were produced from three biological replicates. This has been included in the manuscript.

L163-169: improve general and specific description of the RT-qPCR conditions (volume of cDNA per reaction, primers concentration, temperature, cycles, qPCR final reaction volume…)

R: More details about RT-qPCR reactions have been included.

L181 provide the numbers of each embryo development stage used

R: Numbers have been included as suggested (L204-205).

Figure 2: provide scale bar and y axis identification and unit of the mRNA abundance (do the same for all the other mRNA abundance level figures)

R: Y-axis captions have been included. There is no unit for the quantification of the mRNA abundance. As explained above, the mRNA abundance is relative to the mRNA abundance of the internal control gene (H2A), which is an arbitrary unit.

Reviewer 2 Report

In this article, Da Silva et al. analyze the transcriptional profile of UBs, DUBs, SUMOs and DSUMOS regulators during different stage of pig embryos development. The article is interesting and basically well written but there are some issues that must be solved.

Results in abstract: On first reading, the results section in the abstract is somewhat ambiguous. The authors should try to improve it.

Line 51: is…are.

Line 198 Authors should add here the word blastocyst.

Table 2: Which are the Authors' observations regarding the blastocyst rate of embryos exposed to UV on day 7 of development?

Line 319: DCF 13 into DCAF13.

Line 333: UBC2 into UBA2.

Line 343-346: The Authors should add some references to give strength to this concept.

Fig. 1 panel C: Authors should indicate the caption on the y-axis.

Why did the Authors evaluate the Relative mRNA levels for these genes and the relative mRNA abundance in the other figures? They should eventually standardise the style.

Figure 1 panel A: Authors should add some images with light microscope to appreciate the morphology of the oocytes and embryos at the different stage.

Figure 2: There is no caption on the y-axis. Why relative mRNA abundance and not relative?

Figure 3: Same observations as previously reported.

Figure 4: Again. We have another style to represent the results. Authors should choose one and follow it for all figures. Add the captions on the y axis.

Figure 5: Same observations as for figure 4.

Supplementary Figure 1: Standardise the style as previoulsy reported and add the caption on the y-axis.

In conclusion, the article must be revised.

Author Response

In this article, Da Silva et al. analyze the transcriptional profile of UBs, DUBs, SUMOs and DSUMOS regulators during different stage of pig embryos development. The article is interesting and basically well written but there are some issues that must be solved.

R: We thank the reviewer for highlighting the quality of the manuscript and for the suggestions provided for further improvements.

Results in abstract: On first reading, the results section in the abstract is somewhat ambiguous. The authors should try to improve it.

R: The abstract has been revised and improved.

The structure of the abstract was also changed, as requested by the editor.

Line 51: is…are.

R: Corrected as suggested.

Line 198 Authors should add here the word blastocyst.

R: Included as suggested.

Table 2: Which are the Authors' observations regarding the blastocyst rate of embryos exposed to UV on day 7 of development?

R: D7 blastocysts were exposed to UV and used for analyses 2 h later. We did not observe changes in the blastocyst rate (they were already blastocysts at the time of treatment) at 2 h after treatment. In addition, we did not notice morphological changes after UV exposure.

Line 319: DCF 13 into DCAF13.

R: Corrected as suggested.

Line 333: UBC2 into UBA2.

R: Corrected as suggested.

Line 343-346: The Authors should add some references to give strength to this concept.

R: References have been included.

Fig. 1 panel C: Authors should indicate the caption on the y-axis.

Why did the Authors evaluate the Relative mRNA levels for these genes and the relative mRNA abundance in the other figures? They should eventually standardise the style.

R: Included and modified to abundance.

Figure 1 panel A: Authors should add some images with light microscope to appreciate the morphology of the oocytes and embryos at the different stage.

R: We did not record embryo images at the different stages of development prior to the immunostaining. Embryo morphology after immunostaining and mounting on glass slides is completely lost. Therefore, we do not have light microscopy images of the embryos that were used in this study to include in the manuscript. It is worth highlighting that the main goal for including the pictures was to show the incidence of DNA breaks (gH2AX). We understand that including a set of light microscopy images of embryos will not contribute additional information to this manuscript.

Figure 2: There is no caption on the y-axis. Why relative mRNA abundance and not relative?

R: We used relative mRNA abundance because it is relative to the mRNA abundance of the internal control gene H2A.

We have included the caption for the y-axis as suggested.

Figure 3: Same observations as previously reported.

R: Included as suggested.

Figure 4: Again. We have another style to represent the results. Authors should choose one and follow it for all figures. Add the captions on the y axis.

R: We are not sure about the other style the Reviewer is referring to represent the results.

The results presented in Figures 4 and 5 represent the relative abundance of transcripts in relation to the levels detected in control embryos (not treated with UV) at the same stage of development, i.e., bars below zero indicate that transcripts in the UV-treated embryos were lower than in control embryos, while bars above zero indicate that transcripts in the UV-treated embryos were higher than in control embryos. We believe the format used in Figures 4 and 5 is suitable for the type of results that are presented in those figures.

Captions for the y-axis have been included.

Figure 5: Same observations as for figure 4.

R: Same as above.

Supplementary Figure 1: Standardise the style as previoulsy reported and add the caption on the y-axis.

R: Caption for the y-axis has been included.

In conclusion, the article must be revised.

R: The manuscript has been revised as suggested.

Round 2

Reviewer 1 Report

Authors have addressed all criticism